# SPECTRAL SUBGRAPH LOCALIZATION

## ABSTRACT

Several graph mining problems are based on some variant of the *subgraph isomorphism* problem: Given two graphs, $G$ and $Q$, does $G$ contain a subgraph isomorphic to $Q$? As this problem is **NP**-complete, many methods avoid addressing it explicitly. In this paper, we propose a method that solves the problem by *localizing*, i.e., finding the position of, $Q$ in $G$, by means of an alignment among graph spectra. Finding a node correspondence from $Q$ to $G$ thereafter is relegated to a separate task, as an instance of the *graph alignment* problem. We demonstrate that our spectral approach outperforms a baseline based on the state-of-the-art method for graph alignment in terms of accuracy on real graphs and scales to hundreds of nodes as no other method does.

## 1 INTRODUCTION

Graph analysis tasks frequently require *localizing* a smaller target graph $Q$ within a larger source graph $G$, i.e., finding a subgraph of $G$ that is best aligned with $Q$. This type of problem may appear as *subgraph discovery* (Kuramochi & Karypis, 2001; Bianchini et al., 2018), where we need to find any target graph in $G$, in *subgraph querying* (Katsarou et al., 2015; Sun & Luo, 2019), where we find out whether a target subgraph match exists within a collection of source graphs, or *graph matching* (Zhang & Tong, 2016), where we have to align corresponding nodes across two graphs, potentially of different sizes. Such subgraph localization is of interest in practical applications such as localizing a smaller electronic circuit within a large circuit (Fyrbiak et al., 2019), detecting submolecules in bigger molecules (Najmanovich et al., 2008), and localizing parts of shapes in computational geometry (Rampini et al., 2019). For instance, the task of *subcircuit detection* (Fyrbiak et al., 2019) involves sampling multiple subgraphs and comparing the spectra of their adjacency matrices to that of the query subgraph. Despite the prevalence of the problem, current research has avoided tackling it directly, due to its **NP**-hardness.

In this paper, we propose a novel *spectral* solution to the problem of subgraph localization, built around the notion of identifying the spectrum $\lambda_Q$ of a graph $Q$ within that of another graph $G$. Figure 1 visualizes an instance of the subgraph localization problem by our formulation; we aim to find a function $\delta$ that indicates which nodes in $G$ correspond to $Q$. Our solution effectively recovers both the nodes belonging to the part and the edges that connect the part to the rest of the graph. This problem is an instance of *inverse eigenvalues* problems (Chu & Golub, 2005), the class of problems which aim to reconstruct a matrix from its spectrum.

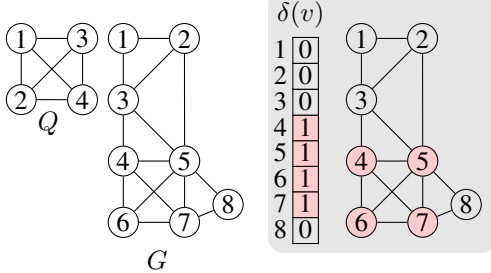

Figure 1: An instance of *subgraph localization* (left) and its solution (right).

Our experimental study demonstrates that our approach tackles the subgraph localization problem more effectively than state-of-the-art neural competitors and showcases its applicability to the real world problem of *subgraph alignment*.

In summary, our contributions are as follows:

- We propose a spectral formulation for the subgraph localization problem (Sec. 4).
- We show that our solution achieves the optimum value under mild conditions (Sec. 3).
- We experimentally validate the effectiveness of our solution on real and synthetic graphs (Sec. 5).

## 2 RELATED WORK

We review related work on five problems related to subgraph localization, namely subgraph isomorphism, subgraph discovery, subgraph querying, subgraph matching, and subgraph localization.

The *subgraph isomorphism* problem is to decide whether a source graph contains a target subgraph and return that exact subgraph in the source. In graph analytics, this problem is mainly solved for very small target subgraphs ($\leq 10$ nodes) and aims at exact matches. Several methods speed up this process by exploiting query specifics, such as patterns in multiple subgraph queries (Duong et al., 2021). By contrast, our method aims at bigger target subgraphs.

In *subgraph discovery*, a target subgraph is not given as input, yet the problem is to identify interesting components of a source graph according to some criteria, as, e.g., those that appear frequently (Kuramochi & Karypis, 2001; 2004), achieve a density threshold (Lee et al., 2010; Qin et al., 2015), or form cliques (Bianchini et al., 2018).

In *subgraph querying*, the goal is to identify all source graphs among a collection that contain a query target subgraph, without necessarily indicating the position of that subgraph within the returned graphs (Katsarou et al., 2015; Sun & Luo, 2019; Sun et al., 2020). A closely related topic is subgraph retrieval, where the goal is to retrieve the most relevant graphs from a graph database, with relevance being measured by some score. In Roy et al. (2022) node embeddings are learned in order to produce a subgraph matching for the computation of the relevance score. In Li et al. (2019), nodes are matched in order to produce a graph similarity score without producing node embeddings as an intermediate step. In both cases, the queries are significantly smaller than ours.

The goal of *subgraph matching* is to match the nodes of a smaller graph to those of a subgraph in a bigger graph via minimizing some error criteria, possibly in the presence of available attribute information. Many methods for graph matching effectively solve a subgraph isomorphism problem, even though they are not specifically designed for this purpose (Zhang & Tong, 2016). Recent work (Lou et al., 2020; Li et al., 2019) employs deep neural models to learn *node embeddings* that are subsequently used for matching.

The problem of *subgraph localization* calls to detect a good fit, by some measure (Skitsas et al., 2023) of a target subgraph within a bigger source graph, without aiming for full isomorphism. This problem has been scarcely studied. A recent application in computer vision (Xu et al., 2020) uses subgraph localization to detect temporal actions, where a graph models actions and the temporal relations between them. However, this model uses edges for temporal aspects and inter-scene relations, and hence does not generalize to arbitrary graphs. An existing spectral solution (Candogan & Chandrasekaran, 2018) is limited to special families of graphs, such as cliques.

## 3 SUBGRAPH LOCALIZATION

All aforementioned problems have in common the search for one graph within another. We study the most generic form of this problem, which corresponds to the problem named *Subgraph localization* in our previous discussion. That is, we aim to identify a subset of the nodes of a graph $G$ corresponding to an input graph $Q$; we do not aim at an exact 1-to-1 correspondence among all graph elements, but to simply detect a set of best matches.

**Problem 1.** *The* subgraph localization *problem for a graph $G = \langle V, E \rangle$, where $V$ is a set of $n$ nodes and $E \subseteq V \times V$ is a set of edges, and a query graph $Q = \langle V_Q, E_Q \rangle$ with $n_Q = |V_Q|$, $n_Q < n$, calls to find a set of nodes $V_S \subset V$, inducing a set of edges $E_S \subset E$, such that $|V_S| = |V_Q|$ and there exists a bijective function $f : V_S \to V_Q$ between the nodes in $V_S$ and those in $V_Q$ such that for each $(i, j) \in E_S$ there exists $(f(i), f(j)) \in E_Q$ and vice versa.*

In many applications, solving subgraph localization, we do not need to explicitly materialize the correspondence function $f$. Such a one-to-one correspondence is not explicitly sought for. Thus, we can eschew recovering an exact $f$ and instead aim at finding an indicator function $\delta : V \to \{0, 1\}$ such that $\delta(v) = 1$, if $v \in V_Q$, and $\delta(v) = 0$ otherwise.

At first glance, finding such an indicator function seems easier than recovering a bijective function $f$. However, even in this identity-function formulation, the problem corresponds to the decision version of the subgraph isomorphism problem, which asks whether a graph $G$ contains a subgraph

isomorphic to another graph $Q$. Thus, the problem is still **NP**-complete. Even so, we further relax our requirements, allowing the function $\delta$ to be a binary version of a continuous real-value function $\mathbf{v} : V \to \mathbb{R}$ on values below a threshold $\tau$:

$$\delta(v) = \begin{cases} 1 & \text{if } \mathbf{v}(v) < \tau \\ 0 & \text{otherwise} \end{cases} \tag{1}$$

This relaxed problem calls to find a real function, or, equivalently, a real vector $\mathbf{v} \in \mathbb{R}^n$, with $n = |V|$, for a known permutation of nodes in the graph. To overcome the requirement for a known node permutation, we consider a permutation-invariant spectral alignment approach reminiscent of the Hamiltonian operator used in shape analysis (Choukroun et al., 2018; Rampini et al., 2019). Before delving into the approach, we introduce the necessary notation.

**Background.** The *adjacency matrix* of graph $G$ with $n$ nodes is a $n \times n$ matrix $\mathbf{A} \in \{0,1\}^{n \times n}$ where $\mathbf{A}_{ij} = 1$ if $(i,j) \in E$, 0 otherwise. The *degree matrix* $\mathbf{D}$ is an $n \times n$ diagonal matrix where each entry $d_{ii} = \sum_{j \neq i} \mathbf{A}_{ij}$ holds the degree of node $i$. The *graph Laplacian matrix* is defined as:

$$\mathbf{L} = \mathbf{D} - \mathbf{A}. \tag{2}$$

The Laplacian matrix of undirected graphs is a positive semi-definite symmetric matrix, hence its eigenvalues $\lambda_1, \ldots, \lambda_n$, are real and non-negative. The *spectrum* $\boldsymbol{\lambda}(\mathbf{M})$ of a matrix $\mathbf{M}$ is the ordered sequence $\lambda_1 \leq \ldots \leq \lambda_n$ of its eigenvalues. Correspondingly, a graph's spectrum is the spectrum of its Laplacian matrix.

## 4 SPECTRAL SUBGRAPH LOCALIZATION

We examine how the presence of a subgraph within a graph affects the graph's spectrum. Spectral theory establishes that the spectrum of a subgraph interlaces with the spectrum of the graph. However, the problem is also non-trivially affected by nodes beside the subgraph. Still, if we could compensate for the effect of nodes other than the subgraph's nodes, the two spectra would be indistinguishable. Following this reasoning, we devise a novel objective for subgraph localization. To that end, we first propose an original connection between subgraph localization and inverse eigenvalue problems with structural constraints (Chu & Golub, 2005).

**Inverse Eigenvalue Problem.** The general *additive inverse eigenvalue problem* (AIEP) is defined as follows:

**Problem 2** (AIEP, Problem 3.6 in Chu & Golub (2005))**.** *Given an $n \times n$ matrix $\mathbf{A}$, a special class of matrices $\mathcal{N}$, and a set of scalars $\{\lambda_{Q_i}\}_{i=1}^k$, find $\mathbf{X} \in \mathcal{N}$ such that $\{\lambda(\mathbf{A} + \mathbf{X})_i\}_{i=1}^k = \{\lambda_{Q_i}\}_{i=1}^k$.*

A vast literature on this problem (see Chu & Golub (2005) and references therein) explores questions regarding the existence of solutions and numerical approximation algorithms for various special classes of matrices $\mathcal{N}$. A common variant of Problem 2 expresses the problem as a least squares problem between the spectra:

$$\min_{\mathbf{X} \in \mathcal{N}} \|\boldsymbol{\lambda}(\mathbf{A} + \mathbf{X}) - \boldsymbol{\lambda}_Q\|^2. \tag{3}$$

In what follows, we establish a connection between the subgraph localization problem (Problem 1) and the additive inverse eigenvalue problem (Problem 2). Under the above formulation, we aim to find a $\mathbf{v}$ that, added to the diagonal of the Laplacian of $G$, renders its first $n_Q$ eigenvalues equal to those of the query graph. In addition to finding $\mathbf{v}$, we aim to remove from $G$ the edges that connect the identified part to the remaining nodes. To the best of our knowledge, this is the first time such a connection has been established, and the first time an AIEP with structural Laplacian constraints is considered.

To devise our solution for subgraph localization, we commence with an intuitive scenario. We assume that $G$ has a number of clearly separated communities, one of which corresponds to the query graph $Q$. A community is defined by a cut, as nodes within the same community are more well connected than nodes across communities. Without loss of generality, assume the graph comprises two distinct communities. In this case, $G$'s Laplacian is a block matrix with two *diagonal*

blocks $\mathbf{L}_{11} \in \mathbb{R}^{n_Q \times n_Q}$ and $\mathbf{L}_{22} \in \mathbb{R}^{(n-n_Q) \times (n-n_Q)}$ and a few entries in the blocks $\mathbf{L}_{12} \in \mathbb{R}^{n_Q \times (n-n_Q)}$ and $\mathbf{L}_{21} \in \mathbb{R}^{(n-n_Q) \times n_Q}$ representing edges across the two communities. The spectra $\boldsymbol{\lambda}(\mathbf{L})$ of $G$ and $\boldsymbol{\lambda}(\mathbf{L}_Q)$ of $Q$ differ on the nodes in $\mathbf{L}_{22}$ and the edges in $\mathbf{L}_{12}$ and $\mathbf{L}_{21}$.

We aim to transform $\mathbf{L}$ into a Hamiltonian Choukroun et al. (2018), defined hereby, to cancel out this difference. A Hamiltonian is an operator $\mathcal{H} = \mathbf{L} + \operatorname{diag}(\mathbf{v})$ where $\mathbf{v} : V \to \mathbb{R}$ is a scalar real-valued function and $\mathbf{L}$ is the Laplacian. The Hamiltonian reduces to the Laplacian if the potential is $\mathbf{0}$. According to (Rampini et al., 2019, Lemma 1), if we add to the diagonal of $\mathbf{L}$ a vector $\mathbf{v}$ having non-zero values, $\mathbf{v}(v) > \tau$, *limited to* nodes in $\mathbf{L}_{22}$, i.e., outside $V_Q$, then eigenvectors corresponding to eigenvalues $\lambda_i < \tau$ of the resulting spectrum $\boldsymbol{\lambda}(\mathbf{L} + \operatorname{diag}(\mathbf{v}))$ will have non-zero values *limited to* the positions corresponding to nodes in $V_Q$, in effect rendering $\boldsymbol{\lambda}(\mathbf{L} + \operatorname{diag}(\mathbf{v}))$ similar to $\boldsymbol{\lambda}(\mathbf{L}_Q)$. Still, the non-zero entries between communities in $\mathbf{L}_{12}, \mathbf{L}_{21}$ affect the spectrum. To cancel that effect, we introduce a *Laplacian editing matrix* that removes the contribution of such edges to the Laplacian of the graph $G$:

$$\mathbf{E} = \begin{bmatrix} -\operatorname{diag}(\mathbf{L}_{12}\mathbf{1}) & \mathbf{L}_{12} \\ \mathbf{L}_{21} & -\operatorname{diag}(\mathbf{L}_{21}\mathbf{1}) \end{bmatrix}$$

where $\mathbf{L}_{12}\mathbf{1}$ (resp. $\mathbf{L}_{21}\mathbf{1}$) corrects the degree of the nodes after removing the edges in $\mathbf{L}_{12}$ (resp. $\mathbf{L}_{21}$). In effect, the corrected Laplacian $\mathbf{L} - \mathbf{E}$ is equivalent to the Laplacian of a graph with two connected components, one of which isomorphic to the query graph $Q$. Thus, the solution $\mathbf{v}$ renders the $|V_Q|$ smallest eigenvalues of the corrected Laplacian indistinguishable from the spectrum of $Q$, $\boldsymbol{\lambda}_Q$, i.e., $\boldsymbol{\lambda}(\mathbf{L} - \mathbf{E} + \operatorname{diag}(\mathbf{v})) = \boldsymbol{\lambda}_Q$, where, with a slight abuse of notation, $\boldsymbol{\lambda}(\mathbf{L} - \mathbf{E} + \operatorname{diag}(\mathbf{v}))$ refers to the $|V_Q|$ smallest eigenvalues of $\mathbf{L} - \mathbf{E} + \operatorname{diag}(\mathbf{v})$.

Since both $\mathbf{v}$ and $\mathbf{E}$ are unknown, we optimize the objective:

$$\min_{\mathbf{v},\mathbf{E}} \; \|\boldsymbol{\lambda}(\mathbf{L} - \mathbf{E} + \operatorname{diag}(\mathbf{v})) - \boldsymbol{\lambda}_Q\|_2^2$$
$$\text{s.t. } \mathbf{E} = \mathbf{E}^\top, \mathbf{E}\mathbf{1} = \mathbf{0}, \; \operatorname{off}(\mathbf{L} - \mathbf{E}) \leq 0, \; \|\mathbf{v}\| = c. \tag{4}$$

This objective is not convex, yet it only depends on the spectrum, for which there exists efficient approximations (Cohen-Steiner et al., 2018); it leads to a solution even if the initial value of $\mathbf{v}$ is a noisy version of the ground truth. As constraints, we postulate that $\mathbf{E}$ should be: (i) symmetric, $\mathbf{E} = \mathbf{E}^\top$; (ii) row- (and, by symmetry, also column-) centered, $\mathbf{E}\mathbf{1} = \mathbf{0}$, with every row summing to 0; and (iii) yielding only non-positive off-diagonal entries $\operatorname{off}(\mathbf{L} - \mathbf{E}) \leq 0$. In addition, we enforce that $\mathbf{v}$ be a point on the surface of a sphere of radius $c$, via the constraint $\|\mathbf{v}\| = c$. Proposition 4.1 provides a sufficient condition on $c$ for the optimality of Equation (4), considering the noiseless case where $G$ exactly contains the subgraph $Q$.

**Proposition 4.1.** *When* $c > \sqrt{n - n_Q} \max(\boldsymbol{\lambda}_Q)$, *the global optimum of Equation* (4) *is obtained at*

$$\mathbf{v} = \begin{cases} 0 & \text{if } v_i \in V_Q \\ \frac{c}{\sqrt{n-n_Q}} & \text{otherwise} \end{cases} \tag{5}$$

*with*

$$\tilde{\mathbf{v}} = \frac{\mathbf{v} - \min(\mathbf{v})}{\max(\mathbf{v}) - \min(\mathbf{v})}, \quad S_{ij} = |\tilde{v}_i - \tilde{v}_j| A_{ij}, \quad \mathbf{E} = \operatorname{diag}(\mathbf{S}\mathbf{1}) - \mathbf{S} \tag{6}$$

*Proof.* Let $\mathbf{E}$ be constructed from Equations 5–6. $\mathbf{L} - \mathbf{E}$ is the Laplacian of a graph composed of two disjoint components, one of which is exactly the component indicated by Equation 5, i.e., the query subgraph $Q$. Then there is a permutation $\boldsymbol{\Pi}$ such that $\boldsymbol{\Pi}\mathbf{L}\boldsymbol{\Pi}^\top$ is a block diagonal matrix with the Laplacian of each component on the diagonal. Without loss of generality, we assume that the Hamiltonian operator attains this block diagonal form:

$$\mathbf{L} - \mathbf{E} + \operatorname{diag}(\mathbf{v}) = \begin{bmatrix} \mathbf{L}_Q & \\ & \mathbf{L}_{\bar{Q}} + \frac{c}{\sqrt{n-n_Q}}\mathbf{1} \end{bmatrix}. \tag{7}$$

When $c$ satisfies the stated condition, the spectrum of the bottom-right block contains only eigenvalues larger than $\max(\boldsymbol{\lambda}_Q)$. It follows that the first $n_Q$ eigenvalues of $\mathbf{L} - \mathbf{E} + \operatorname{diag}(\mathbf{v})$ are exactly those of $\mathbf{L}_Q$, rendering the objective of Equation 4 equal to zero. $\qquad\square$

In effect, by Proposition 4.1, we can recover the optimal solution if $\mathbf{v}$ is appropriately normalized and $c$ is no less than a certain value. We exploit this result in Section 4.2 to design our algorithm by numerical optimization. We first introduce a regularization term.

**Regularization.** The objective in Equation 4 does not prevent $\mathbf{v}$ from taking arbitrary values. However, since $\mathbf{L} - \mathbf{E}$ has two connected components, $\mathbf{v}$ plays a role similar to that of Fiedler's vector in the minimization of the normalized cut (Shi & Malik, 2000). This observation leads us to the *spectral regularization* $\mathbf{v}^\top (\mathbf{L} - \mathbf{E})\mathbf{v}$ that exhorts $\mathbf{v}$ to take values in the null-space of $\mathbf{L} - \mathbf{E}$. In other words, the spectral regularizer drives $\mathbf{v}$ to be a stepwise function. We combine the spectral regularization with our objective as follows:

$$\min_{\mathbf{v},\mathbf{E}} \underbrace{\|\boldsymbol{\lambda}(\mathbf{L} - \mathbf{E} + \mathrm{diag}(\mathbf{v})) - \boldsymbol{\lambda}_Q\|_2^2}_{\text{Data term}} + \mu \underbrace{\mathbf{v}^\top (\mathbf{L} - \mathbf{E})\, \mathbf{v}}_{\text{Spectral regularizer}} \tag{8}$$

$$\text{s.t. } \mathbf{E} = \mathbf{E}^\top, \mathbf{E}\mathbf{1} = \mathbf{0}, \ \mathrm{off}(\mathbf{L}-\mathbf{E}) \leq 0, \ \|\mathbf{v}\| = c$$

where $\mu \geq 0$ is a regularization coefficient.

**Corollary.** *Proposition 4.1 applies also with the spectral regularization term in Equation* (8).

*Proof.* Let $\mathbf{E}$ be constructed from Equations (5)–(6). $\mathbf{L} - \mathbf{E}$ is the Laplacian of a graph composed of two disjoint components, one of which is exactly indicated by Equation (5), i.e., the query subgraph $Q$. Then $\mathbf{v}$ in Equation (5) belongs to the null-space of $\mathbf{L} - \mathbf{E}$, rendering the regularization term 0, hence Equation (5) also provides the global minimum of Equation (8). □

### 4.1 LOCALIZING DISCONNECTED SUBGRAPHS

A special case of subgraph localization is that of a graph with a number of connected components, one of which corresponds to the query graph $Q$. In this case the editing matrix $\mathbf{E} = \mathbf{0}$, leading to the simpler objective:

$$\min_{\mathbf{v}} \|\boldsymbol{\lambda}(\mathbf{L} + \mathrm{diag}(\mathbf{v})) - \boldsymbol{\lambda}_Q\|_2^2 + \mu \mathbf{v}^\top \mathbf{L}\mathbf{v} \quad \text{s.t. } \|\mathbf{v}\| = c. \tag{9}$$

### 4.2 NUMERICAL OPTIMIZATION

We exploit Proposition 4.1 to craft a numerical procedure that minimizes the objective in Equation (4), collaterally optimizing for $\mathbf{E}$ and $\mathbf{v}$. In the first iteration $q = 0$, we initialize $\mathbf{E}_q = \mathbf{0}$. In iteration $q + 1$ we minimize $f(\mathbf{v}, \mathbf{E}_q) = \boldsymbol{\lambda}(\mathbf{L} - \mathbf{E}_q + \mathrm{diag}(\mathbf{v})) - \boldsymbol{\lambda}_Q\|_2^2 + \mu \mathbf{v}^\top (\mathbf{L} - \mathbf{E}_q)\, \mathbf{v}$ for $\mathbf{v}$ given $\mathbf{E}_q$:

$$\mathbf{v}_{q+1} = \arg \min_{\mathbf{v}:\|\mathbf{v}\|=c} f(\mathbf{v}, \mathbf{E}_q), \tag{10}$$

via *projected gradient descent*, until convergence; an iteration $k + 1$ of projected gradient descent performs the step:

$$\mathbf{x}_{k+1} = \mathbf{x}_{k+1} - \alpha \nabla_{\mathbf{v}} f(\mathbf{v}, \mathbf{E}_q)$$
$$\mathbf{v}_{k+1} = c \frac{\mathbf{x}_k}{\|\mathbf{x}_k\|}, \tag{11}$$

where $\alpha > 0$ regulates the learning rate. The gradient $\nabla_{\mathbf{v}}$ for Equation 11 requires a *differentiable eigendecomposition*, which is achievable by extant methods (Wang et al., 2019).

We subsequently update $\mathbf{E}$ according to:

$$\tilde{\mathbf{v}} = \frac{\mathbf{v}_q - \min(\mathbf{v}_q)}{\max(\mathbf{v}_q) - \min(\mathbf{v}_q)}, \tag{12}$$

$$S_{ij} = |\tilde{v}_i - \tilde{v}_j| A_{ij}, \tag{13}$$

$$\mathbf{E}_{q+1} = \mathrm{diag}(\mathbf{S}\mathbf{1}) - \mathbf{S}. \tag{14}$$

We obtain a threshold $\tau$ of the indicator function $\delta(\mathbf{v})$ in Equation 1 for the nodes comprising the subgraph by splitting the elements of $\mathbf{v}$ into two clusters minimizing sum-of-squares error from the mean (i.e., optimizing the $k$-means objective in one dimension) and compute the matrix $\mathbf{E}$ from this thresholded $\mathbf{v}$ by Equations 12–14.

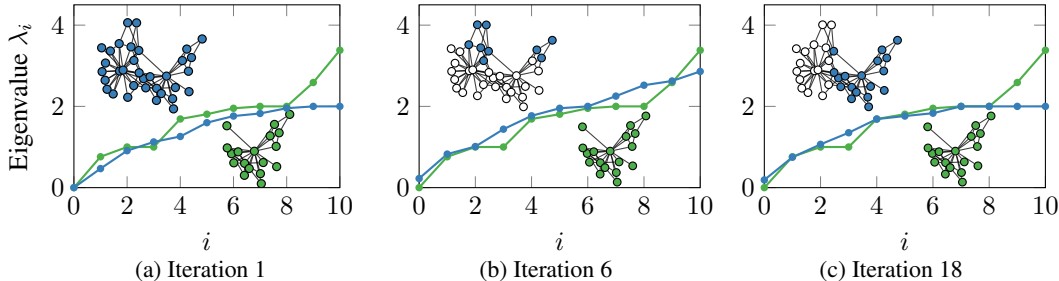

| (a) Iteration 1 | (b) Iteration 6 | (c) Iteration 18 |

Figure 2: Alignment of the spectrum $\boldsymbol{\lambda}_Q$ of $Q$ and the part of the spectrum $\boldsymbol{\lambda}(\mathbf{L} - \mathbf{E} + \mathrm{diag}(\mathbf{v}))$ of $G$ corresponding to $Q$ at 1, 6, and 18 iterations. As two spectra progressively converge, especially in smaller eigenvalues, $Q$ is correctly localized in $G$.

**The SSL algorithm.** We eventually present our *Spectral Subgraph Localization* (SSL) algorithm (Algorithm 1 in the supplementary material) for Problem 1. SSL takes as input the adjacency matrix $\mathbf{A}$ of the full graph $G$ and the spectrum of a query subgraph, and returns the vector $\mathbf{v}$ and the threshold $\tau$ of the indicator function $\delta$; it additionally requires some hyperparameters, such as the number of outer iterations $\texttt{maxiter}_{\texttt{out}}$, the number of inner iterations $\texttt{maxiter}_{\texttt{in}}$, the learning rate $\alpha$, and the regularization coefficient $\mu$. We empirically found that the number of iterations and the learning rate do not significantly affect results across datasets if chosen within some range; we report those ranges and recommended values in Table 1 in the supplementary material. On the other hand, the regularization coefficient $\mu$ in Equation 8 requires tuning for each dataset. We thus first normalize the value of $\mu$ by $c^2$ to remove the dependency on $\mathbf{v}$'s magnitude and then perform grid search on a range of values for $\mu$ to select an appropriate value.

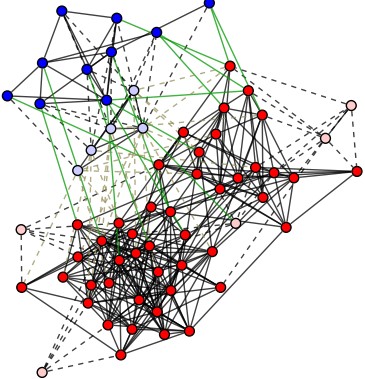

Figure 3: Example SSL result: ground-truth subgraph $Q$ in blue; remaining nodes of $G$ in red.

The optimization process alternates the projected gradient optimization in Equation 11 and the update of $\mathbf{E}$ using Equations 12–14 until it converges or reaches the maximum number of iterations $\texttt{maxiter}_{\texttt{out}}$. Figure 2 illustrates the solution's progressive convergence through iterations, while Figure 3 shows an example result.

### 4.3 COMPLEXITY ANALYSIS

We derive the worst-case time complexity of the algorithm in the number of nodes $n$ in the graph $G$. The eigendecomposition in Equation 11 takes $\mathcal{O}(n^3)$ per iteration; the computation in Equation 14 takes $\mathcal{O}(n^2)$ for the matrix-vector multiplication; 1-D k-means in Line 9 takes $\mathcal{O}(n)$ with the best algorithm (Grønlund et al., 2017). In effect, the total time is $\mathcal{O}(\texttt{maxiter}_{\texttt{out}} \cdot (\texttt{maxiter}_{\texttt{in}} * n^3 + n^2 + n))$, where the $\mathcal{O}(n^3)$ term dominates. However, as $\mathbf{L} - \mathbf{E} + \mathrm{diag}(\mathbf{v})$ is a graph's Laplacian, its spectrum can be efficiently approximated through sampling (Cohen-Steiner et al., 2018).

## 5 EXPERIMENTS

Here we empirically evaluate our method, SSL, on a number of datasets and against several hypotheses. Our evaluation aims to answer the following questions:

**(Q1)** Do the regularization term and the constraint $\|\mathbf{v}\| = c$ in Equation (8) help the localization?

**(Q2)** How does the conductance of the part corresponding to $Q$ affect the quality of localization and how does SSL fare against state-of-the-art methods for graph alignment?

**(Q3)** What kind of graphs are challenging for SSL and why?

## 5.1 EXPERIMENT DESIGN

The code and data are available at `https://anonymous.4open.science/r/SSL-F39A`.

**Hyperparameters.** Unless stated otherwise, we choose $\texttt{maxiter}_{\texttt{out}} = 3$, $\texttt{maxiter}_{\texttt{in}} = 500$, $a_{\texttt{tol}} = 10^{-5}$ and $\alpha = 0.02$. Regarding the regularization coefficient, we select $\mu = 0.2$ through grid search. This choice achieves good accuracy across datasets and conductance levels.

**Datasets.** We evaluate SSL on the three real-world graphs from Rossi & Ahmed (2015) and two synthetic graphs generated by the Erdős-Renyi (ER) and Barabasi-Albert (BA) models. The data characteristics are described in the supplementary material. Additionally, we generate graphs with community structure using the stochastic block model (SBM) (Holland et al., 1983).

**Choosing $Q$.** Given a number $k$, we generate a query workload of size $V_Q = k$ from a real-world graph $G$ to evaluate our subgraph localization method as follows.

1. Randomly select a node $u$, add it to $V_Q$ and place all its neighbors into a set $N$.
2. Randomly select a node $u'$ from $N$, add it to $V_Q$, place in $N$ all its neighbors not in $V_Q$.
3. Repeat the previous step until $|V_Q| = k$.
4. Set $Q$ as the subgraph induced by $V_Q$ in $G$.

For graphs generated by the stochastic block model, we set $Q$ as the smallest community.

**Quality measure.** To evaluate performance in a manner independent of subgraph size, we use **Balanced Accuracy (BA)** (Brodersen et al., 2010); given the query graph $V_Q$ and the subgraph $V_S$ returned by a localization algorithm, balanced accuracy $\mathrm{BA}(\mathbf{v}) = \frac{1}{2}\left(\frac{|V_Q \cap V_S|}{|V_Q|} + \frac{|\neg V_Q \cap \neg V_S|}{|\neg V_Q|}\right)$ is the arithmetic mean of sensitivity (or recall) and specificity.

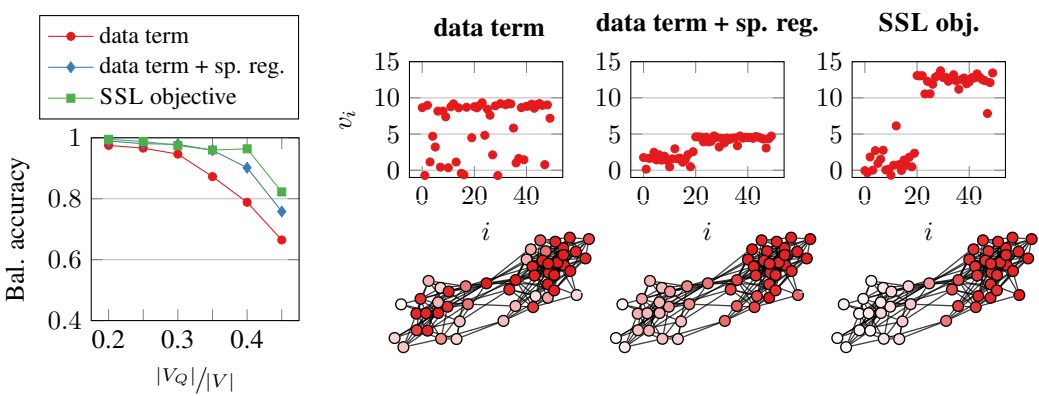

(a) Balanced Accuracy          (b) $\mathbf{v}$ values in ascending order and corresponding graph nodes.

Figure 4: Variants of the objective function; SSL's objective achieves the highest accuracy.

## 5.2 ABLATION STUDY

We commence our study by examining how the terms in SSL's objective function (Equation 8) affect the result. Recall that the objective function consists of: (1) the **data term**, that drives the alignment between the spectrum of the part and that of the query, (2) a **spectral regularization** term that exhorts $\mathbf{v}$ to be in the null space of $\mathbf{L} - \mathbf{E}$ and (3) the **sphere constraint** that enforces a constant norm on the potential $\mathbf{v}$. To study the contribution of each term on the results, we compare SSL against two variants thereof:

1. A method only optimizing the data term $\|\boldsymbol{\lambda}(\mathbf{L} - \mathbf{E} + \mathrm{diag}(\mathbf{v})) - \boldsymbol{\lambda}_Q\|_2^2$.
2. A method optimizing a linear combination of the data term $\|\boldsymbol{\lambda}(\mathbf{L} - \mathbf{E} + \mathrm{diag}(\mathbf{v})) - \boldsymbol{\lambda}_Q\|_2^2$ and the spectral regularization $\mathbf{v}^\top (\mathbf{L} - \mathbf{E}) \mathbf{v}$, without a sphere constraint.

We experiment on graphs with $|V| = 200$ nodes sampled form the stochastic block model, letting the size of the query subgraph increase from 20% of the graph to 45%. Figure 4a reports on the results of this ablation study in terms of average balanced accuracy over 5 sampled graphs for each subgraph size. Unsurprisingly, the optimization of the data term yields the worst results, although the method performs well on small subgraphs. Still, the addition of the spectral regularizer and sphere constraint enhances the results up to 20% accuracy. For small subgraphs the sphere constraint brings only marginal gains compared to the spectral regularization. On the other hand, on large query subgraphs, the sphere constraint boosts the accuracy by an additional 8%.

To further corroborate these results, Figure 4b shows an example of how the terms impact the potential $\mathbf{v}$, on a 40-node graph sampled from the SBM with two communities with 20 nodes each; the query graph is one of the two communities. Ideally, we would like to obtain a $\mathbf{v}$ clearly separating values between the part corresponding to the query graph and the rest. In that case, we say that $\mathbf{v}$ forms a step function. The optimization of the *data term* (left chart) alone leads to no clear separation between the two parts. Introducing the *spectral regularization* (middle chart) yields a result closer to a step function, though some nodes are incorrectly assigned to the part. Finally, the full objective in Equation 8 produces to a clearly separated potential vector $\mathbf{v}$. Visualizing the mapping of this potential to the graph $G$, we clearly recognize the part $G_S$ as the light-colored nodes.

## 5.3 COMPETING METHODS

Here we assess our method against previous work. To the best of our knowledge, no extant unsupervised method is capable to answer localization queries in graphs with more than 15 nodes (Roy et al., 2022). Therefore, we compare SSL to the nearest feasible competitor, namely the state-of-the-art method for unsupervised graph alignment, CONE (Chen et al., 2020). To set up CONE so that it detects subgraphs, we inject in the query nodes with degree 0, so that the size of the query $Q$ corresponds to that of the graph $G$, i.e., $|V_Q = V|$. We extract the ensuing localization vector as the matches of query nodes in $G$ with the default hyper-parameter settings.

Figure 5 and Figure 6 present the average BA of 10 randomly generated connected subgraphs as a function of *conductance*, $\Phi(V_Q) = \frac{\sum_{i \in V_Q, j \notin V_Q} A_{ij}}{\min\left(\sum_{i \in V_Q, j \in V} A_{ij}, \sum_{i \notin V_Q, j \in V} A_{ij}\right)}$, i.e., the ratio between the size of the cut among query $Q$ and graph $G$ and the minimum number of edges among the two resulting partitions. A graph's *minimum conductance* is associated (Cheeger, 2015) to its *algebraic connectivity*, i.e., second smallest eigenvalue $\lambda_2$ (Fiedler, 1973). A larger conductance denotes more edges between the query subgraph and the rest of the graph, thus a harder subgraph localization instance. We use query subgraphs corresponding to 10%, 20% and 30% of the full graph size.

The results in Figure 5 show that SSL effectively localizes the query in real graphs. Accuracy gradually increases as the conductance approaches $\Phi(V_Q) = 0$, finally settling at 100% accuracy on all datasets, when the query is disconnected. In the Malaria dataset, we note a more abrupt increase. The performance of SSL is always comparable to, and most often exceeds, that of CONE. While performance drops as conductance grows, in real applications we would aim at detecting interesting subgraphs that exhibit distinguishable structures, such as social communities. Such subgraphs deviate substantially from both random and complete subgraphs. We model these nontrivial connectivity patterns by a lower conductance. As conductance increases, the subgraph progressively becomes merged into other nodes, hence SSL cannot discriminate it.

The results in Figure 6 show that SSL consistently outperforms CONE on synthetic graphs. As with real graphs, we observe a gradual accuracy increase as the graph becomes progressively disconnected. Notably, on ER graphs, SSL succeeds even at high conductance values ($> 0.6$).

**Impact of the graph's spectrum.** To better understand the performance of SSL on different graphs, we look at it under the lens of the graph's spectrum. Figure 7 shows the spectra of the real (Figure 5) and synthetic graphs (Figure 6) in our experiments, normalized in the range $\left[0, \frac{\lambda_n - \lambda_2}{\lambda_n}\right]$. First, we observe that the spectrum of synthetic graphs exhibits a gradual increase and a small difference between $\lambda_2$ and the maximum eigenvalue $\lambda_n$. By the *Generalized Cheeger's inequality* (Lee et al., 2010) the $k^{\text{th}}$-order conductance, $\min_{V_1, V_2, ..., V_k} \max\{\Phi(V_i) : i = 1, 2, ..., k\}$, is related to the $k^{\text{th}}$ eigenvalue. We conclude that, under gradual eigenvalue growth, the presence or absence of one edge does not affect the spectrum significantly, hence the projected gradient descent in SSL

gracefully retrieves a good solution. On the other hand, the spectra of real graphs in our experiments exhibit an abrupt divergence between $\lambda_2$ and higher eigenvalues, indicating that a single edge may significantly affect the spectrum, rendering the task of projected gradient descent more challenging. In effect, SSL performs better as the gap between $\lambda_2$ and the rest of the eigenvalues decreases.

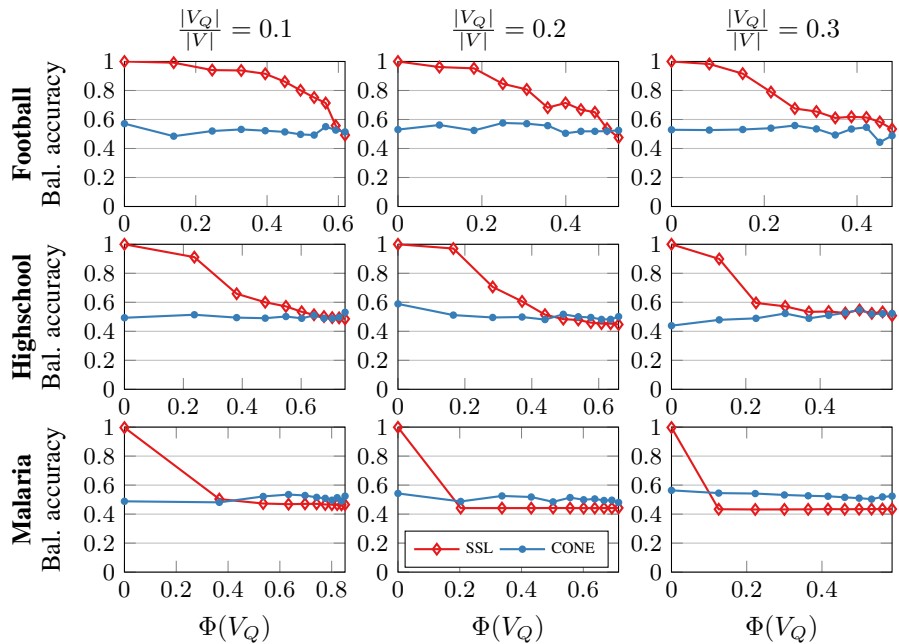

Figure 5: Accuracy vs. conductance between query subgraph $Q$ and graph $G$, real graphs.

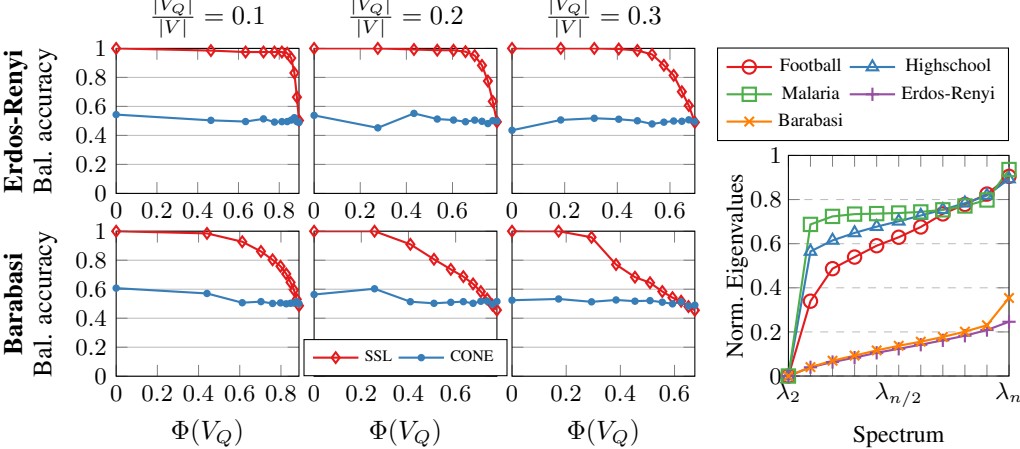

Figure 6: Accuracy vs. conductance, ER and BA graphs.                Figure 7: Graph spectra.

## 6 CONCLUSION

We studied the challenging problem of subgraph localization, which calls to find a set of nodes in a larger graph that best corresponds to a given subgraph. We devised a novel spectral solution that identifies the query match by adding a penalty to the Laplacian matrix so as to obtain a spectrum similar to that of the query graph. This novel approach requires solving a non-convex, non-smooth problem for which we devised a numerical method. Our results demonstrate that our spectral method localizes query subgraphs more effectively than a baseline based on the state-of-the-art method for graph alignment. To our knowledge, this is the first endeavor in effective subgraph localization that can handle graphs of any size in the order of magnitude of hundreds of nodes.

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

# A  SUPPLEMENTARY MATERIAL

We implemented SSL in Python 3.6 and ran experiments on an 8-core Intel Core i7-8565U machine with 16GB RAM. Code and data available at `https://anonymous.4open.science/r/SSL-F39A`.

## A.1  HYPERPARAMETER SETTING

We calibrated SSL using grid search on the hyperparameters $\mathtt{maxiter_{in}}$, $\mathtt{maxiter_{out}}$, $a_{\mathtt{tol}}$, and $\alpha$. The range of the tested hyperparameters is depicted in Table 1. We observe no significant difference in the hyperparameters in different datasets, vindicating the robustness of our method. Surprisingly, we observe the same robustness for the regularization parameter $\mu$.

| Parameter | Value/range | Description |
|---|---|---|
| $\mathtt{maxiter_{in}}$ | 500–1000 | number of inner iterations |
| $\mathtt{maxiter_{out}}$ | 3–5 | number of outer iterations |
| $a_{\mathtt{tol}}$ | $10^{-5}$ | loss tolerance |
| $\alpha$ | 0.02 | gradient step size |

Table 1: SSL hyperparameters and default values.

## A.2  DATASET DESCRIPTION

Table 2 and Table 2 presents the characteristics of the datasets used in the experimental evaluations in the number of nodes $V$, edges $E$, network type, and parameters.

| Dataset | Source | $|V|$ | $|E|$ | Network type |
|---|---|---|---|---|
| Football | Girvan & Newman (2002) | 115 | 613 | Contact |
| Malaria | Larremore et al. (2013) | 306 | 9042 | Biological |
| HighSchool | Fournet & Barrat (2014) | 327 | 5 818 | Proximity |

Table 2: Real graphs used in our evaluation: number of nodes $|V|$, number of vertices $|E|$, and graph type.

| Dataset | Source | $|V|$ | $|E|$ | Parameters |
|---|---|---|---|---|
| Barabási–Albert | Barabási & Albert (1999) | 200 | 5907 | $m_{new} = 50$ |
| Erdős–Rényi | Erdős et al. (1960) | 200 | 8185 | $p_{new} = 0.5$ |

Table 3: Synthetic graphs used in our evaluation: number of nodes $|V|$, number of vertices $|E|$, number of edges to attach from a new node to existing nodes $m_{new}$, edge creation probability $p_{new}$.

## A.3  CHALLENGING CASES

In this section, we investigate some examples of challenging cases for SSL, illustrated in Figures 8 and 9. In the results of Figure 8, we observe that the spectrum of the query graph and that of the detected subgraph are well aligned. However, the localized subgraph deviates substantially from the ground truth. Similarly, in the results of Figure 9, while SSL does not perfectly align the two spectra, it yields a correlated spectrum. Nevertheless, SSL detects a subgraph comprising nodes that are only connected by one edge. In both cases, the challenge arises from the sensitivity of the spectrum at weakly connected parts of the graph. Changing the Laplacian in such parts by adding $\mathbf{v}$ has a larger impact on the spectrum than changing the Laplacian in a well-connected neighborhood. These types of graphs force the optimization process into a local optimum, as the optimizer has a large incentive to separate these weakly connected parts of the graph.

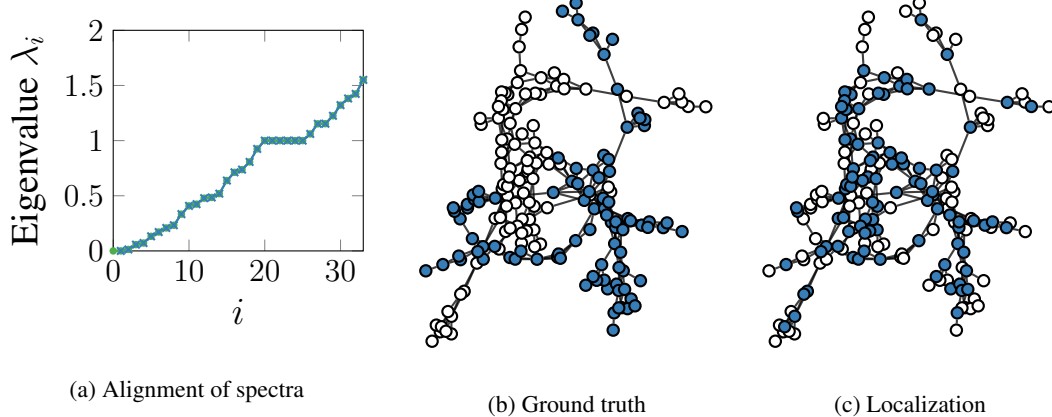

(a) Alignment of spectra          (b) Ground truth          (c) Localization

Figure 8: Alignment of the spectrum $\boldsymbol{\lambda}_Q$ of $Q$ and the corresponding part of the spectrum $\boldsymbol{\lambda}(\mathbf{L} - \mathbf{E} + \text{diag}(\mathbf{v}))$ of $G$ after convergence, ground truth $V_S$ (blue) and $V \backslash V_S$ (white), and corresponding localization by SSL; while the spectra are perfectly aligned, the detected subgraph is not the ground truth. The depicted graph is a protein-protein interaction network from the D&D dataset Dobson & Doig (2003).

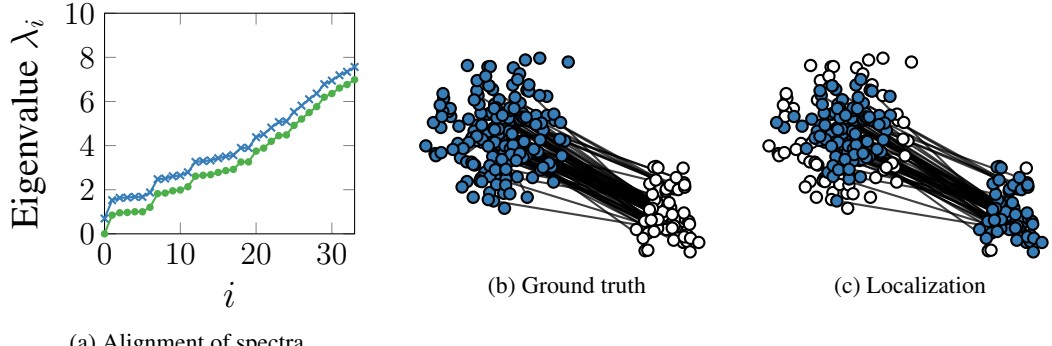

(b) Ground truth          (c) Localization

(a) Alignment of spectra

Figure 9: Alignment of the spectrum $\boldsymbol{\lambda}_Q$ of $Q$ and the corresponding part of the spectrum $\boldsymbol{\lambda}(\mathbf{L} - \mathbf{E} + \text{diag}(\mathbf{v}))$ of $G$ after convergence, ground truth $V_S$ (blue) and $V \backslash V_S$ (white), and corresponding localization by SSL; while the spectra are well aligned, the detected subgraph is not the ground truth; the detected subgraph coincides with nodes that have degree 0 or 1. The depicted graph is **arenas** Kunegis (2013).

## A.4  SSL PSEUDOCODE

Algorithm 1 shows the psedocode of SSL presented in Section 3. The algorithm is easy to implement in a few lines of code.

**Algorithm 1** SSL

---

**Input:** $\mathbf{A}$ adjacency matrix of the full graph; $\boldsymbol{\lambda}_Q$ spectrum of the query subgraph.

**Params:** $\mu$ regularization coefficient; $a_{\mathtt{tol}}$ loss tolerance; $\alpha$ gradient step size; $\mathtt{maxiter_{in}}$ maximum number of inner iterations; $\mathtt{maxiter_{out}}$ maximum number of outer iterations

**Output:** Vector $\mathbf{v}$, threshold $\tau$

1: $\mathbf{L} \leftarrow \mathbf{D} - \mathbf{A}$
2: $\mathtt{loss} \leftarrow \infty$
3: $c \leftarrow 2\sqrt{n - n_q} \max(\boldsymbol{\lambda}_Q)$
4: $\mathbf{v}_0 \leftarrow \frac{c}{|V|}\mathbf{1}$
5: $\mathbf{E}_0 \leftarrow \mathbf{0}$
6: **while** $q \leq \mathtt{maxiter_{out}}$ **and** $\mathtt{loss} \geq a_{\mathtt{tol}}$ **do**
        // Compute $\mathbf{v}_{q+1}$ by iterating (11) $\mathtt{maxiter_{in}}$
7:      $\mathbf{v}_{q+1} \leftarrow \arg\min_{\mathbf{v}:\|\mathbf{v}\|=c} f(\mathbf{v}, \mathbf{E}_q)$
        // Update $\mathbf{E}_{q+1}$ via (12))–(14)
8:      $\mathbf{E}_{q+1} \leftarrow \mathtt{E\_from\_v}(\mathbf{v}_{q+1})$
        // Update the threshold $\tau$
9:      $\tau \leftarrow \mathtt{k\_means\_1d}(\mathbf{v}_{q+1})$
10:      $\mathtt{loss} \leftarrow f(\mathbf{v}_\tau, \mathbf{E}_\tau)$
11:      $q \leftarrow q + 1$
12: **return** $\mathbf{v}_q, \tau$

---

