# OpenReview forum: "Spectral Subgraph Localization"
_ICLR.cc/2023/Conference — Submitted to ICLR 2023_

### Official Review · Reviewer_aKY2 · 2022-10-25

**Confidence:** 3
**Correctness:** 4
**Technical Novelty And Significance:** 2
**Empirical Novelty And Significance:** 4
**Recommendation:** 3

**Clarity, Quality, Novelty And Reproducibility:**

The paper is written in a very clean way. I really appreciate the nice manner in which the presentation progresses. I would have preferred if the authors had stated the main theoretical results of the paper upfront, but nevertheless this does not take away much from the clarity of the paper.

**Strength And Weaknesses:**

The major strength of the paper comes from the experimental evaluation of SSL.No unsupervised method is able to handle query graphs Q on more than 15 vertices to the best of what authors know. Thus, I find it particularly impressive that SSL can handle graphs up to 200 nodes (thanks to their choice of a different spectral formulation).

On the other hand, the obvious weakness is the lack of theoretical results to go alongside. While I do not insist on theoretical results being present in this same submission, there is another weakness that the authors themselves bring up (but do not address). This weakness has to do with trying to understand what kind of query graphs Q are hard for SSL and why? The current version does not offer any insight on this problem and this is something I would have really liked to see.

**Summary Of The Paper:**

The paper under review studies the problem of graph localization. This can be thought of as a problem belonging to the venerable family of problems falling under the subgraph isomorphism problem and its variants. The paper under review defines the subgraph localization problem as the following: You are given a query graph Q along with its spectra. You are also given a host graph G. The challenge is to find a subset of vertices in G which induces a copy of Q. As stated this is a NP-Hard problem. The results of the paper suggest that the spectral subgraph localization approach (the one developed in the paper) can find a copy of Q in G if G admits a community-like structure and Q is one of the communities in G.

Theoretically, Spectral Subgraph Localization (SSL) solves an inverse eigenvalue problem (which is non-convex). This inverse problem is formulated as an optimization problem where you try to minimize the l_2^2 difference between the eigenvalues of Q and eigenvalues of an "edited" Laplacian. You minimize this difference over all the edits and the minimizer is deemed to reveal the set Q.

The authors have supporting experiments in the paper. These experiments suggest that the SSL approach recovers 80% fraction of the vertices when G comes from the Stochastic Block Model.

**Summary Of The Review:**

Overall, the experimental results in this work are fairly intriguing. While there are not enough theoretical results to accompany the experimental findings, I think the paper can spur some theoretical investigation in the analysis of SSL (or its variants). For this reason alone, I recommend an accept verdict.

========================================

(EDIT: added later)

After a detailed discussion with my fellow reviewers and a bit of pouring over the experiments section a little more carefully, I have to say that I must eat my words from before.

In particular, now I vote to reject the paper. As is evident, the paper does not have any ammunition on the theory side. What was gunning for the paper in my mind earlier was the experimental side. Let me make a few remarks about that now.

It is clear from Section 5.2 that this set of plots is used to back the SSL objective as the winning horse among some of its minor variants. To me, this is not much of a value add to the technical prowess of the paper. After my fellow reviewers pointed out to me, I agree the SSL algorithm only works for well-separable graphs.

Given these concerns, I now change my vote to reject the paper.

---

> ### Author Response · Authors · 2022-11-17
> **Response to review aKY2**
>
> Thank you for the time taken to review this paper and provide thorough feedback. Please find answers below:
>
> _What kind of query graphs Q are hard for SSL and why?_
>
> **A**: Query graphs with multiple similar occurrences in the target graph are harder to localize spectrally, as their multiplicity misleads the optimizer.
>
> _state main theoretical results of the paper upfront_
>
> **A**: Our main theoretical result is the spectral formulation of the subgraph localization problem as an inverse eigenvalue problem, as stated in Section 1.

---

### Official Review · Reviewer_rJzq · 2022-10-27

**Confidence:** 4
**Correctness:** 4
**Technical Novelty And Significance:** 3
**Empirical Novelty And Significance:** 3
**Recommendation:** 8

**Clarity, Quality, Novelty And Reproducibility:**

The paper is nicely written. Proofs for the claims have been provided. Originality of the work is strong.

**Strength And Weaknesses:**

A major strength of the paper is in the discovery of the connection between the SGL problem and the additive inverse eigen value problem. A major weakness lies in the accuracy obtained. For example, when |V_Q|/|V| is 0.5, the balanced accuracy drops to around 0.65.


**Summary Of The Paper:**

This paper addresses the problem of subgraph localization (SGL). The goal is to identify a good fit of a target subgraph within a larger source graph. This problem is related to the subgraph isomorphism problem and is NP-hard. The authors establish a connection between the SGL problem and the additive inverse eigen value problem. This connection leads to a spectral based algorithm called SSL for solving the SGL problem. This algorithm identifies the nodes and edges in the source graph corresponding to the target subgraph. Experimental results are encouraging.

**Summary Of The Review:**

This paper addresses the problem of subgraph localization (SGL). The authors establish a connection between the SGL problem and the additive inverse eigen value problem. This connection leads to a spectral based algorithm called SSL for solving the SGL problem. This algorithm identifies the nodes and edges in the source graph corresponding to the target subgraph. Experimental results are encouraging. The authors could focus on improving the balanced accuracy especially when |V_Q| is large.

---

> ### Author Response · Authors · 2022-11-17
> **Response to review by rJzq**
>
> Thank you for the time you took to review this paper and provide thorough feedback. Please find an answer below:
>
> _when |V_Q|/|V| is 0.5, the balanced accuracy drops to around 0.65._
>
> **A**: Indeed, the problem becomes harder as the ratio |V_Q|/|V| grows.

---

### Official Review · Reviewer_oSHA · 2022-10-30

**Confidence:** 4
**Correctness:** 3
**Technical Novelty And Significance:** 2
**Empirical Novelty And Significance:** 2
**Recommendation:** 3

**Clarity, Quality, Novelty And Reproducibility:**

As described above, the presentation is somewhat confusing. It is heavily dependent on previous work, which is fine but it does need extra work to make it cleaner. For example, not many knowledgable readers may not know what a Hamiltonian transform would mean.

The writeup is easy to read but it will be very helpful to have a crisper problem definition and a separation from the overall approach. For instance the structure can be -- problem definition, intuition/simplification, and then finally the exact algorithm for the general case.

Further, as I mentioned above, why can't we separate the computation of the matrix X (that can be added to matrix A) and then compute E and diag(v), by decomposing X.



**Strength And Weaknesses:**

The key difference between the subgraph isomorphism problem and the alignment problem is that one doesn’t focus on computing the isomorphism function explicitly but instead only determine the indicator function.

The paper proposes to “relax” this problem further to a continuous version where the goal is to compute, instead of the indicator function, a real-valued function on the node set such that the indicator function can be retrieved by applying a thresholding on the real-valued function.

Q: Given the real-valued function, there are a linear number of threshold function possible. So how exactly is this real-valued version a relaxation of the indicator version of the problem.


The central idea behind their approach is to connect subgraph localization to another well-studied problem known as inverse eigenvalue problem. Essentially, in this problem, we are given base matrix and a set of target eigenvalues, and the goal is to compute a matrix (from a family of matrices) that can be added to the base matrix such that the resulting matrix has eigenvalues equal to the target values. One of the several approaches to solve is problem is formulating it as a least squares problem.

The key idea of the work is to characterize the addendum matrix (to the base matrix) as the difference of another matrix E and a diagonal matrix whose diagonal values correspond to the real-valued indicator function. The matrix E is constrained as a symmetric matrix satisfying some additional properties.

The loss function is then minimized by using projected gradient descent. There is no convergence shown but assumed that a max-iters parameter can be specified in practice to limit the runtime.

Q: Why can’t we first compute the matrix X that can be added to the based matrix A and then decompose X to compute matrices E and delta matrix? It appears that the first problem is solved by state-of-the-art approaches and the latter is a much simpler problem. Perhaps the benefit of this could be easier convergence proof?


Pros:

1.	The characterization of the problem is indeed interesting, and the paper does present some good theoretical and experimental results.

Cons:
1.	Presentation is extremely confusing. For example, it is not clear right now, if there is any difference between computing the delta indicator function vs the real-valued function. It is indeed easier to compute the real-valued function, but the indicator function can easily be retrieved from it by thresholding – then why is relaxation not simple an aspect of the solution (instead of defining it as a new problem variant).
2.	There are no guarantees whatsoever on the quality. The well-separability assumption is not mild as claimed and I think even the experimental results seem to show that clearly (as the conductance increases the loss of the approach in fact is worse than the baseline, in figure 5).
3.	There is no guarantee on the convergence. The paper claims that in the experiment they see quick convergence, then I think it should be possible to show some convergence on the theoretical side. Otherwise, just characterizing the loss function and applying project gradient descent is just too weak from a theoretical point of view.
4.	Finally, I am not convinced of the applicability of the approach itself to that of subgraph detection. The measure of quality is that of balanced accuracy but I am not sure if it would mean anything in reality. For instance, if I obtain a subgraph with 0.9 accuracy, it could be a completely wrong subgraph. A better measure might be – how many times were you able completely identify the correct subgraph (upto isomorphism). Otherwise, I am sure how one can use this.


**Summary Of The Paper:**

The paper studies the subgraph localization problem -- given a target graph and a source graph, compute the alignment of the target graph in the source graph. The problem of alignment is closely related to that of detecting isomorphic subgraph in a source graph. The goal here is to compute an indicator function that identifies a subset of nodes in the source graph that can be isomorphically mapped to the target subgraph.


Key contributions:

1.	Propose a new spectral approach for the alignment problem.
2.	Under “mild” assumptions, specifically on the separability of the source graph, prove that their approach yields optimum value under a certain loss function.
3.	Experimental validation of their approach.


**Summary Of The Review:**

I think the paper has some promise but needs to be significantly rewritten to make it cleaner. The idea of decomposing X into E and diag(v), subject to some relevant constraints on E, is where I see the claimed novelty. But as discussed, I not completely convinced of the effectiveness of the approach (both from a theoretical side and practice).

==== update after the rebuttal ======

I would like to thank the authors for their feedback. Post reading other reviews + feedback + discussions with reviewers, I think while the problem is somewhat interesting, there are some keys issues that need to be addressed:

1. The theory is simply non-existent. So there are no guarantees on the quality of the algorithm. While, the suggested simple approach may indeed be "bad"  as the authors, it is not clear why (perhaps that can be a point of exploration, simplification of the algorithm or proof that you cannot simplify)
2. More importantly, the experimental results are not convincing enough. Indeed, the graph constructed seem to be tailored to ensure well separability and therefore the algorithm has a definite advantage over state of the art. From the results, in fact, it is easy to deduce that as the graphs become close to real, the algorithm looses any advantage over SOTA ..

For the above reasons, I am inclined to reject the current paper.

---

> ### Author Response · Authors · 2022-11-17
> **Reponse to review by oSHA**
>
> Thank you for the time you took to thoroughly review this paper. Please find our responses below:
>
> _Given the real-valued function, there are a linear number of threshold functions possible. So how exactly is this real-valued version a relaxation of the indicator version of the problem._
>
> **A**: By relaxation, we return continuous real-valued, rather than binary solutions. In the binary case, the problem is NP-complete, while in the relaxed version, it becomes tractable.
>
> _Why can’t we compute the matrix $\mathbf{X}$ that can be added to the base matrix $\mathbf{A}$? It appears that this problem is solved by state-of-the-art approaches._
>
> **A**: We initially formulated the problem as the general inverse eigenvalue problem in Equation 3. However, we observed poor results due to the unconstrained $\mathbf{X}$. Our problem requires that $\mathbf{L+X}$ be a _Laplacian matrix_, and that $\mathbf{X}$ has a steep transition to discriminate query nodes from the rest of the graph. These requirements render the direct application of solutions for the inverse eigenvalue problem insufficient.
>
> _Is there any difference between computing the delta indicator function vs the real-valued function? It is indeed easier to compute the real-valued function, but the indicator function can easily be retrieved from it by thresholding – then why is relaxation not simply an aspect of the solution?_
>
> **A**: Yes, there is a difference, as by relaxing the indicator from binary to continuous, we are solving an arguably easier, albeit related, _continuous_ optimization problem. The solution amounts to solving that easier problem by an off-the-shelf projected-gradient-descent solver and thresholding by 1-D k-means.
>
> _The well-separability assumption is not mild as claimed and I think even the experimental results seem to show that clearly (as conductance increases the loss of the approach is worse than the baseline, in Figure 5)._
>
> **A**: We introduce the well-separability assumption as a mild one in the context of detecting _interesting_ subgraphs that exhibit distinguishable structures, such as social communities or proteins with a specific biological function. Such subgraphs deviate substantially from both random and complete subgraphs as they present nontrivial connectivity patterns. Still, as conductance increases, the subgraph progressively becomes merged into other nodes, hence SSL cannot discriminate it.
>
> _The paper claims that in the experiment they see quick convergence, then I think it should be possible to show some convergence on the theoretical side._
>
> **A**:  We have indeed overloaded the verb “converge”. The sentence "the projected gradient descent in SSL gracefully **converges** at a good solution" should have been "the projected gradient descent in SSL gracefully **retrieves** a good solution". We amend this lapse in the revised version.
>
> _if I obtain a subgraph with 0.9 accuracy, it could be a completely wrong subgraph. A better measure might be – how many times were you able to completely identify the correct subgraph (up to isomorphism)._
>
> **A**: We aim to localize the subgraph, rather than to find exact node-to-node correspondences, as in the subgraph isomorphism problem. A 0.9-accuracy solution is still relevant. For instance, with a query graph of size 100 and a graph of size 1000, ba=0.9 would mean we identified 82% of the query graph correctly, since 0.9 = (82/100 + 882/900)/2. An isomorphism-based measure aims to identify **exact** matches, which may simply not exist.
>
> _Knowledgeable readers may not know what a Hamiltonian transform means._
>
> **A**: The Hamiltonian transformation is described in Page 4, adding a vector to, and subtracting an editing matrix from, the diagonal of the Laplacian, leading to the Hamiltonian operator in Equation (7).
>
> _the structure can be -- problem definition, intuition/simplification, and then finally the exact algorithm._
>
> **A**: Indeed, Section 3 provides the problem definition. Section 4 offers the intuition/simplification, and then finally Section 4.2 presents the exact algorithm.

---

### Author Response · Authors · 2022-11-18
**Revised version**

As requested, we have added the following elements in the revised paper in blue color:

1. Page 4: a clarification on the Hamiltonian transformation and its components.
2. Page 8: a discussion on the effect of conductance on the results in Figure 5.
3. Page 9: a correction of the overloaded verb “converges to” to “retrieves”.

---

### Decision · Program_Chairs · 2023-01-20

**Decision:**

Reject

**Justification For Why Not Higher Score:**

Please see above

**Justification For Why Not Lower Score:**

N/A

**Metareview: Summary, Strengths And Weaknesses:**

We carefully discussed this paper and we think (1) the theoretical results are not quite up-to-bar, (2) the experiments seem to only work for well-separable graphs and so perhaps not on real data; though they use real data to generate some of their datasets, they combine this with a synthetic procedure to decrease the conductance so it is not clear how practically valuable the results are.

**Summary Of Ac-Reviewer Meeting:**

Points (1) and (2) above were the main focus of our discussion. We all agreed there was a lack of solid theory so focused on the value of the experiments and ultimately came to the conclusion in (2) above.